# INFER: A NEURAL-SYMBOLIC MODEL FOR EXTRAPOLATION REASONING ON TEMPORAL KNOWLEDGE GRAPH

**Ningyuan Li**[1], **Haihong E**[1]*, **Tianyu Yao**[1], **Tianyi Hu**[1], **Yuhan Li**[1], **Haoran Luo**[1],
**Meina Song**[1], **Yifan Zhu**[1]
[1]Beijing University of Posts and Telecommunications

## ABSTRACT

Temporal Knowledge Graph(TKG) serves as an efficacious way to store dynamic facts in real-world. Extrapolation reasoning on TKGs, which aims at predicting possible future events, has attracted consistent research interest. Recently, some rule-based methods have been proposed, which are considered more interpretable compared with embedding-based methods. Existing rule-based methods apply rules through path matching or subgraph extraction, which falls short in inference ability and suffers from missing facts in TKGs. Besides, during rule application period, these methods consider the standing of facts as a binary 0 or 1 problem and ignores the validity as well as frequency of historical facts under temporal settings. In this paper, by designing a novel paradigm for rule application, we propose IN-FER, a neural-symbolic model for TKG extrapolation. With the introduction of Temporal Validity Function, INFER firstly considers the frequency and validity of historical facts and extends the truth value of facts into continuous real number to better adapt for temporal settings. INFER builds Temporal Weight Matrices with a pre-trained static KG embedding model to enhance its inference ability. Moreover, to facilitates potential integration with existing embedding-based methods, INFER adopts a rule projection module which enables it apply rules through conducting matrices operation on GPU. This feature also improves the efficiency of rule application. Experimental results show that INFER achieves state-of-the-art performance on various TKG datasets and significantly outperforms existing rule-based models on our modified, more sparse TKG datasets, which demonstrates the superiority of our model in inference ability.

## 1 INTRODUCTION

Knowledge Graphs(KGs) play key roles in multiple downstream applications (Hildebrandt et al., 2019; Lan & Jiang, 2020), which store knowledge in the form of triples $(s, r, o)$ representing subject entity $s$ and object entity $o$ are linked by the relation $r$. Traditional KGs appear to be static snapshots of real-world facts. However, in practice, facts evolve over time and may not be true in a perpetual way. Thus, TKGs are proposed to model the dynamic properties of facts in which each fact is represented as a quadruple $(s, r, o, t)$ where $t$ denotes the timestamp of the fact.

Extrapolation reasoning on TKGs focuses on predicting potential facts at future timestamps based on the observed historical facts. Some effective embedding-based models including RE-Net (Jin et al., 2019), CyGNet (Zhu et al., 2021a), RE-GCN (Li et al., 2021), CENET (Xu et al., 2022) etc. have been proposed for TKG extrapolation, which embed entities and relations into vectors and utilize neural networks to capture structural information, temporal dependency and historical information. Embedding-based models exhibit strong learning and reasoning abilities and yield substantial performance. However, due to the "black box problem" of neural networks, embedding-based methods are deemed to be lacking interpretablity and reliability. Recently, several rule-based methods like TLogic (Liu et al., 2022) and TR-Rules (Li et al., 2023) have been proposed for better

---

*Corresponding author.

**Rule:** $(X, Unsatisfied\_With, Y, T_1) \bigwedge (Y, Trade\_With, Z, T_2) \rightarrow (X, Plays\_For, Z, T)$

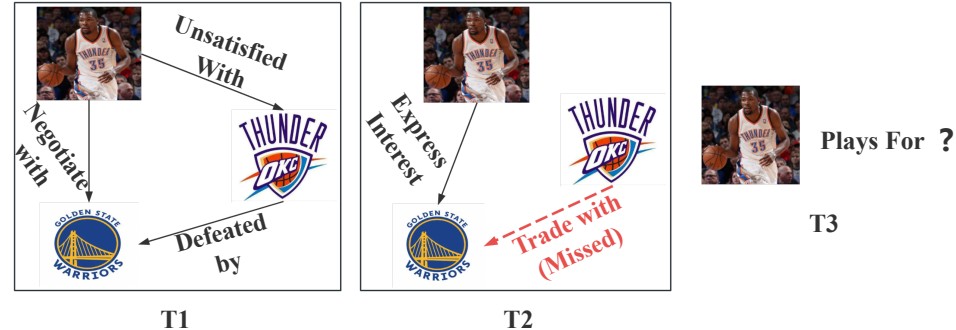

Figure 1: Illustration of how missing facts in TKGs affect the rule application procedure.

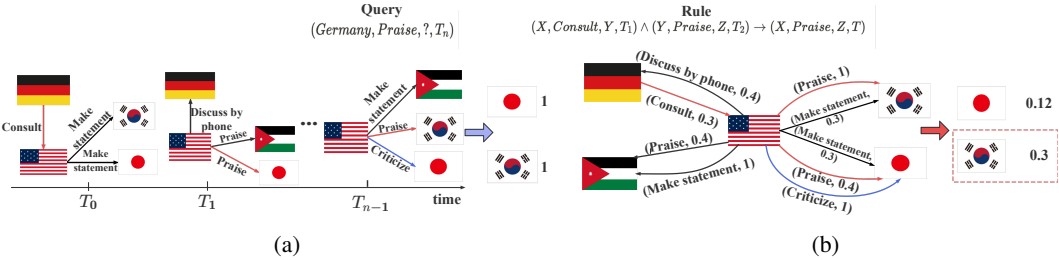

Figure 2: (a) Illustration of TKGs and previous rule-based inference. (b) Our transformed weighted graphs and the performed temporal validity aware inference.

interpretability. These models can generate human-readable rules and provide visible reasoning chains, which also achieve competitive performance.

Nevertheless, when applying rules, previous rule-based methods consider the standing of facts as a binary 0 or 1 problem (have been true or not) for path matching, which ignores the validity as well as frequency of historical facts under temporal settings. Although previous models try to add bias in the score function to increase the score of candidates derived by recent edges or paths, it does not completely solve this issue especially when rule length is greater than one.

In Figure 2 (a), we can see a typical TKG and the process of rule application of previous work. The red edges denote matched rule body instances. As we can see, Germany consulted the U.S. at $T_0$ and the U.S. praised Japan and Korea at $T_1$ and $T_{n-1}$ respectively. Given the query and the rule, previous models will consider both entities correct and give them the same score. However, intuitively, these two "Praise" edges should not be treated equally because of the time difference and in fact, the U.S. even criticized Japan at $T_{n-1}$.

Additionally, existing rule-based models apply rules through purely symbolic ways such as: path matching or subgraph extractions which do not have the ability to infer missing facts in TKGs and make the application of rules conditioned on the quality of datasets. Because of the incompleteness of TKGs, we believe this drawback brings a bottleneck to previous rule-based methods. In Figure 1, we can see that although the model mines a plausible rule and it could have derived the correct answer, the missing facts at T2 makes the reasoning failed. However, existing rule-based methods need to traverse and match rule body instances in TKGs on CPUs, which brings efficiency bottlenecks and hinders integration with embedding-based neural methods.

To address these issues, we propose INFER, a neural-symbolic model for TKG extrapolation, which introduces a novel rule application paradigm. INFER first learns rules in TKGs with an existing random walk based algorithm. Then, in our paradigm, INFER builds Temporal Weight Matrices with the help of static KG embedding models, which store the probability of each possible fact. Notably, this procedure enables the assignment of initial approximate probabilities to even missing facts, thereby enhancing the inference capabilities of our model.

We argue that the binary truth value utilized by previous works for modeling the standing of facts is not suitable enough for temporal settings. To capture the effects of the frequency and validity of historical facts, we design a Temporal Validity Function that maps the frequency and the time span since the last appearance of a fact into a continuous real number. When it comes to a new timestamp, the temporal validity function is used to update the temporal weight matrices which store the temporal validity of each historical fact and probability of potential unseen facts. If we consider the temporal weight matrices as adjacent matrices, this process continuously transforms the graphs of historical facts into a weighted graph. Figure 2 (b) shows an example of the weighted graph computed by INFER.

As for rule application, INFER introduces a rule projection module. The rule projection module computes scores of all candidates through conducting efficient matrices calculation on GPU according to the mined rules rather than costly path matching on graph. Hence, INFER is able to apply rules more efficiently than previous models can and makes the application of longer rules affordable. This design facilitates potential integration with existing embedding-based reasoning methods. Moreover, INFER is able to display the scores of all entities at each intermediate step of rule application, which maintains the interpretability of rule-based methods. Code is available at `https://github.com/JasonLee-22/INFER`.

Our contributions can be summarized as follows:

- We argue that continuous value rather than binary value reflects the truth value of historical facts more precisely under temporal settings. We introduce a Temporal Validity Function which considers the time span and frequency of facts together to calculate continuous truth values of temporal facts.

- We propose INFER, a neural-symbolic model for TKG extrapolation, which adopts a novel paradigm for rule application. INFER constructs and dynamically updates temporal weight matrices to augment the ability to infer missing facts and capture historical clues. By performing calculations on these matrices on GPU, INFER enables more efficient rule application, making it possible to combine with embedding-based methods and maintain the interpretablity meanwhile.

- Experimental results on multiple datasets show that INFER achieves state-of-the-art performance. And INFER also gives more robust performance compared with existing rule-based models on our modified datasets where a certain proportion of facts at each timestamp are removed.

## 2 RELATED WORK

Embedding-based reasoning is a mainstream methodology for TKG extrapolation, which embeds entities, relations and timestamps into low-dimensional vector space and then utilizes neural networks to learn the structural information, temporal dependency and historical patterns in TKGs. RE-NET (Jin et al., 2019) uses RNN and RGCN to learn representations of subgraphs which encode structural and sequential information. CyGNet (Zhu et al., 2021a) introduces copy-generation mechanism to make the model focus more on the repetitive historical facts. TITer (Sun et al., 2021) uses reinforcement learning to search candidates in historical TKGs, which is guided by a reward based on Dirichlet distribution. CENET (Xu et al., 2022) argues that unseen facts are also critical in prediction and introduces contrastive learning to identify whether the current timestamp relies more on historical facts or non-historical facts. DaeMon (Dong et al., 2023) models the temporal path information between a pair of nodes based on the NBFNet (Zhu et al., 2021b) graph neural network and introduces a memory passing strategy to update the path information.

For better interpretability, xERTE (Han et al., 2021) performs reasoning on query-based subgraphs and captures structural as well as temporal information. TLogic (Liu et al., 2022) is the first rule-based TKG extrapolation model which mines rules in TKGs through temporal random walk and applies rules to predict future events via matching rule body instances. TR-Rules (Li et al., 2023) proposes a new and appropriate algorithm to calculate the confidence of rules under temporal settings. It also manages to mine and apply acyclic rules in TKGs which are proven to be effective. ALRE-IR (Mei et al., 2022) combines embeddings with rules by learning rule representations, which is further utilized for confidence assessment. Although embeddings are introduced, it is just used for estimating confidence of rules and ALRE-IR still can not model the frequency and time validity of historical facts.

## 3 PRELIMINARIES

### 3.1 TKG EXTRAPOLATION

In this paper, $\mathcal{E}$, $\mathcal{R}$ and $\mathcal{T}$ denote the set of all the entities, relations and timestamps respectively. We use $|\mathcal{E}|$ and $|\mathcal{R}|$ to represent the number of entities and relations. TKG can be viewed as a sequence of graphs i.e. $\mathcal{G} = \{G_1...G_n\}$. Each graph $G_i$ consists of all the facts that occur at timestamp $t_i$, i.e. $G_i = \{(s, r, o, t)|t = t_i\}$ where $s, o \in \mathcal{E}$, $r \in \mathcal{R}$ and $t_0 \leq t_i \leq t_n$. Interpolation Reasoning on TKGs attempts to answer queries whose timestamps range from $t_0$ to $t_n$. However, TKG extrapolation aims at predicting the answer for a future query $(s, r, ?, t')$ or $(?, r, o, t')$ where $t_n < t'$, based on the historical facts $\{G_1...G_n\}$.

### 3.2 TEMPORAL RULES

Cyclic rules under temporal settings can be defined as:

$$(E_1, r_h, E_{n+1}, T_{n+1}) \leftarrow \bigwedge_i^n (E_i, r_{bi}, E_{i+1}, T_i)$$

where $E_i$ and $T_i$ denote variables of entity and timestamps and $r_i \in \mathcal{R}$ represents a specific relation. The left part of the rule is called the rule head, while the right part is called the rule body. In some cases, it is restricted that the same entity need to appear repetitively so that the rule body is satisfied. We call this phenomenon variable constraint. Corresponding examples and illustration can be found in Appendix A. During the symbolic rule learning procedure, each rule will be associated with corresponding confidence which is calculated by sampling on the graphs. Normally, rules do not always hold and sometimes are violated, thus the confidence of a rule represents the probability of this rule being correct.

## 4 METHOD

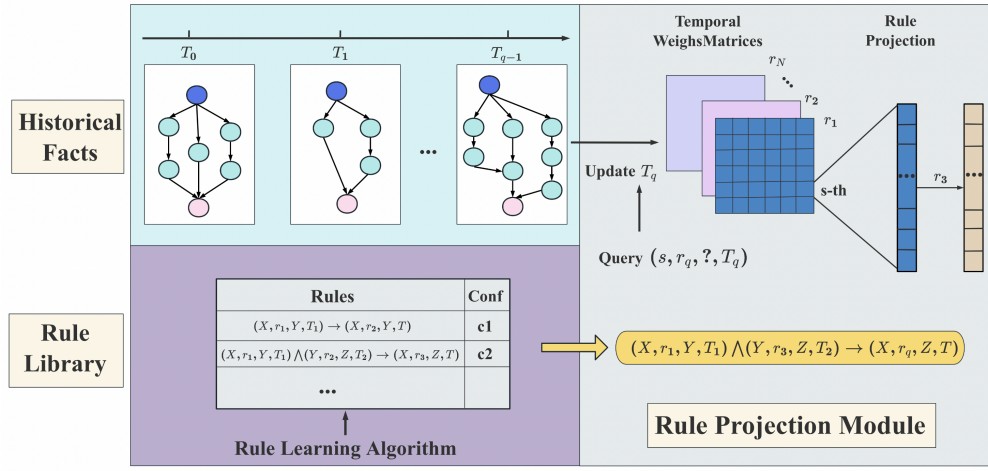

Figure 3: Overview of INFER. INFER first mines rules in TKGs with existing rule learning algorithm and builds temporal weight matrices with a pre-trained static KG embedding model. When it comes a new timestamp, INFER update the temporal weight matrices according to the historical facts. Then the rule projection module selects corresponding rules and applies rules by performing matrices operation for rule projection.

### 4.1 OVERVIEW

Figure 3 illustrates the overall framework of INFER. Inspired by QTO (Bai et al., 2023), INFER builds temporal weight matrices with a static KG embedding model which assigns probabilities to all possible facts to enable the model to infer missing facts during rule application. INFER first uses the rule learning algorithm of previous symbolic methods to build a rule library. Then INFER introduces a Temporal Validity Function which maps the frequency and time span of a historical fact into a continuous value between 0 and 1. The Temporal Validity Function is used to update the temporal weight matrices at each new timestamp. Given a query, the rule projection module selects corresponding matrices or vectors for projection according to the relevant rules stored in rule library and finally gives the scores of all candidates. In most cases, more than one rule can be applied for a query, INFER leverages Noisy-OR to aggregate scores given by multiple rules for final prediction.

### 4.2 RULE LEARNING

We first mine temporal rules in TKGs and calculates their confidence to build a rule library for rule application afterwards. We do not extensively search for the optimal rule learning algorithm (including rule confidence calculation) for INFER but adopt the rule learning algorithm from TR-Rules (Li et al., 2023) across all experiments. We believe better rule learning algorithm with high-quality rules and more accurate confidence will further improve the performance of INFER. We leave the exploration of other rule learning methods such as the newly released (Huang et al., 2024) for future work. We only utilize cyclic rules for extrapolation reasoning and filter out those acyclic rules and store all the mined cyclic rules with their corresponding confidence.

### 4.3 RULE PROJECTION MODULE

#### 4.3.1 TEMPORAL WEIGHT MATRICES

We pre-train a static KG embedding model by ignoring the timestamp in each quadruple. In INFER, we select ComplEx (Trouillon et al., 2016) as the static KG embedding model. While other KG embedding models can be used as alternatives, we believe that a suitable KG embedding model can improve the overall performance of INFER, although it is not a definitive factor.

Then, for each relation $r \in \mathcal{R}$, we define a temporal weight matrix $M_r \in \mathbf{R}^{|\mathcal{E}|*|\mathcal{E}|}$. The i-th row in $M_r$ records the weights of the edges of type $r$ that link entity $e_i$ to all other entities, while the j-th column denotes the weights of edges linking each entity to entity $e_j$ with $r$. This intuition guides the rule projection in the following section. The pre-trained KG embedding model can score each possible non-temporal facts. However, the scores are real numbers, we need to re-scale the scores to [0,1] resulting in a valid probability. Given head entity $e_i \in \mathcal{E}$ and relation $r$, we apply the softmax function to the scores of all possible candidates and the corresponding values are stored in the temporal weight matrix of relation $r$:

$$M_r(i,j) = \frac{exp(f(e_i, r, e_j))}{\sum_{1 \leq k \leq |\mathcal{E}|} exp(f(e_i, r, e_k))} \tag{1}$$

where $f$ is the score function of the KG embedding model, $i$ and $j$ are the indices of entities.

There are two reasons why we utilize static embedding models instead of temporal models. The first one is that if we use temporal embedding models then the size of the matrix for relation $r$ is $M_r \in \mathbf{R}^{|\mathcal{E}|*|\mathcal{E}|*|\mathcal{T}|}$ which costs too much space. The other reason is that since temporal models score the same fact at various timestamps differently and we do not consider specific timestamps during application, static embedding models which capture the overall structural information is enough for INFER. For resource savings, following QTO (Bai et al., 2023), we set a threshold value to filter out those values smaller than the threshold.

#### 4.3.2 TEMPORAL VALIDITY FUNCTION

The rule application algorithm of previous symbolic methods is to match instanced rule bodies in observed graphs. This process can be viewed as first assigning binary truth values to historical facts or assigning binary weights to edges in graphs: **1** for facts have appeared, **0** for unseen facts, i.e. no

edges. Then the model traverses the graph to search whether there are instanced paths matching the rule body and finally gives binary results of whether the instanced rule heads hold true. However, the temporal validity and frequency of facts cannot be modeled in this way. To be specific, the facts that occurred recently should not be treated as equally as the facts that held true long ago (Liu et al., 2022). And facts that frequently appeared should be more significant and trustworthy.

Hence, we attempt to extend the truth value to continuous real number via considering time validity and frequency of historical facts and propose the following Temporal Validity Function:

$$V(s, r, o, t_c) = \frac{1}{\sqrt{t_c - t_{last}}} + \lambda \cdot \sqrt{\mathcal{F}} \cdot M_r(s, o) \tag{2}$$

$$\lambda = \sigma(\frac{\sqrt{\mathcal{F}}}{\tau}) - \frac{1}{2} \tag{3}$$

where $t_c$ denotes the current timestamp, $t_{last}$ represents the latest timestamp when fact $(s, r, o)$ stands, $\mathcal{F}$ is the frequency of $(s, r, o)$ counted from 0 to $t_{c-1}$, $\sigma(\cdot)$ is the sigmoid function, $\tau$ is a hyper-parameter for smoothing and $M_r$ is the temporal weight matrix introduced above. It should be noticed that $t_{last}$ should satisfy $t_{last} \leq t_{c-1}$, since we can not observe facts concurring with the query.

As we can see that the Temporal Validity Function consists of two parts: the time span and the frequency. The time span part which is the first item in the formula means that the credibility of a fact weakens as time goes on as long as it does not recur. The square root ensures the attenuation is neither too fast nor insensitive for measuring the validity. The motivation of designing the second item which is the frequency part is that for those frequently appearing facts, we believe they are more consistent and robust. The frequency serves to amplify the temporal weight of a fact when it is recurrent. While $\lambda$ is the weight of the frequency part ranging from 0 to 0.5. It gets bigger as the fact repeats more times in history, which helps to alleviate the attenuation of time span part when a frequent fact does not occur temporarily.

When it comes to a new timestamp $t \in \mathcal{T}$, given $(s, r, o) \in H_t$ which is a historical fact, we use the above function to update the temporal weight matrices:

$$M_r(s, o) = \mathbf{min}\{V(s, r, o, t), 1\} \tag{4}$$

To a certain extent, this design of Temporal Validity Function is an empirical result. We provide more details and analysis about this part in Appendix D.

### 4.3.3 RULE PROJECTION

When answering a query $(s, r_q, ?, t')$, INFER finds relevant rules $S_{r_q}$ with respect to $r_q$ from the rule library. For each rule in $S_{r_q}$, we traverse its rule body $\bigwedge_i^n (E_i, R_i, E_{i+1}, T_i)$ to get all the relations in order so that the temporal causality is maintained $\{R_1 \otimes R_2... \otimes R_n\}$. Then, inspired by the multi-hop reasoning operation proposed in QTO (Bai et al., 2023), the rule is projected iteratively:

$$\mathbf{Ans_i} = \begin{cases} M_{R_i}[s, :] & i = 1 \\ \max_{col}(\mathbf{Ans_{i-1}^T}_{\cdots \times |\mathcal{E}|} \odot M_{R_i}) & 2 \leq i \leq n \end{cases} \tag{5}$$

Firstly, we select the s-th row of $M_{R_1}$ which denotes the scores of each entity serving as object entity in facts with subject entity $s$ and relation $rl_1$. If the rule length is more than 1, we transpose the row vector and repeat it along the row direction for $|\mathcal{E}|$ times which results in a matrix of $|\mathcal{E}| \times |\mathcal{E}|$. Then we calculate the Hadamard product of the obtained matrix and the corresponding temporal weight matrix. Finally, we take the maximal value of each column to obtain a new row vector.

This procedure will be repeated $n - 1$ times and the final row vector is the scores of all candidates given by the rule. The row vector can be viewed as a fuzzy set representing the scores of all entities at the intermediate step of reasoning. Then, the Hadamard product computes the scores of all entities with a fuzzy set rather than a specific entity, thus giving a matrix. The column-wise max operator selects the highest value among all possible paths for each candidate. This process can be considered as another way to traverse and search in graphs which is based on operating on the generalized adjacent matrices of weighted graphs.

---

**Algorithm 1** Pseudocode of Rule Projection Without Variable Constraints

---

**Require:** $(s, r_q, ?, T_q)$: Query; $\{R_1, R_2...R_n\}$: Corresponding rule bodies; $M_{\mathcal{R}}$: Temporal Weight Matrices;

1: set $\mathbf{Ans} = \mathbf{M_{R_1}}[s, :]$    # Select the s-th row of $\mathbf{M_{R_1}}$
2: **for** $i$ **in** $(2, n+1)$ **do**
3:    $\mathbf{Ans} = \mathbf{Ans^T}.\mathbf{repeat}(1, |\mathcal{E}|)$    #Repeat the transposed vector along row direction
4:    $\mathbf{Ans} = \mathbf{Ans} \odot \mathbf{M_{R_i}}$    #Hadamard Product
5:    $\mathbf{Ans} = \mathbf{max}(\mathbf{Ans}, dim = -1)$    #Column-wise max operation giving a row vector
    **return Ans**

---

However, in some cases, there are variable constraints in rules longer than 1. As we can see in (6), the instantiated rule bodies must comply with the constraint that the entity serving as $E_2$ must be the same.

$$(E_1, demands\_for\_change, E_2, T_4) \leftarrow (E_1, intent\_to\_change, E_2, T_1)$$
$$\wedge (E_2, Host\_a\_visit, E_3, T_2) \wedge (E_3, Make\_a\_Visit, E_2, T_3) \quad (6)$$

In INFER, we also design the projection algorithms for these rules. The detailed pseudocode and illustration are provided in Appendix A. Notably, when processing rules with variable constraints, additional indexing operations and calculations are required, leading to a delay in inference speed.

## 4.4 INFERENCE

Normally, more than one rule can be used to answer a query. In INFER, we aggregate the scores of multiple rules with the Noisy-OR function (Meilicke et al., 2019a):

$$score(q) = \mathbf{1} - \prod_{\mathbf{Ans_r} \in R_q} (\mathbf{1} - c_r \cdot \mathbf{Ans_r}) \quad (7)$$

where $q$ denotes the query, $R_q$ represents the set of all scores given by rules, $c_r$ is the confidence of the rule and $\mathbf{1}$ is an all-one vector. Apart from the rule confidence, the Noisy-OR aggregation is equivalent to the union of the fuzzy sets generated by the rules.

In some cases, there might be no proper rules for answering the query $q$. In INFER, for a query $(s_n, r_n, ?, t_n)$ which does not have corresponding rules, we directly predict it based on the temporal weight matrix by considering the query itself as a rule of length 1. For those queries that have corresponding rules, we also make each of the queries itself as a rule of length 1 and use it for prediction. Because we consider the prediction based on this "generated rule" represents the distribution implied by pure temporal information and frequent interactions in historical facts. We assign the average confidence of all rules to this "generated rule".

## 5 EXPERIMENTS

### 5.1 EXPERIMENTAL SETUP

**Datasets**   We evaluate INFER on five TKG datasets: ICEWS14, ICEWS18, ICEWS0515, YAGO and WIKI. The first three datasets are subsets of Integrated Crisis Early Warning System (Boschee et al., 2015). These datasets record international facts that occurred in 2014, 2018 and from 2005 to 2015 respectively. Table 3 in Appendix B shows the statistics of these five datasets. To satisfy the extrapolation reasoning settings, facts are sorted in ascending order based on timestamps and then split into train, valid and test.

**Metrics and Implementation Details**   We utilize mean reciprocal rank (MRR) and Hits@k as metrics to evaluate INFER. $MRR = avg(\frac{1}{rank_i})$ and $Hits@k = \frac{1}{N}\sum_i \mathbf{I}(rank_i < k)$, where $\mathbf{I}$ is the indicator function. Higher values of both metrics indicate better performance of models. We adopt the time-aware filtering protocol proposed in xERTE (Han et al., 2021), which turns out to be more reasonable in temporal settings. We implement INFER with PyTorch and all experiments are conducted on a single 48GB NVIDIA A40 GPU. More details about implementation are reported in Appedndix E.

| | ICEWS14 | | | | ICEWS05-15 | | | | ICEWS18 | | | |
| Model | MRR | Hits@1 | Hits@3 | Hits@10 | MRR | Hits@1 | Hits@3 | Hits@10 | MRR | Hits@1 | Hits@3 | Hits@10 |
|---|---|---|---|---|---|---|---|---|---|---|---|---|
| Dismult | 27.67 | 18.16 | 31.15 | 46.96 | 28.73 | 19.33 | 32.19 | 47.54 | 10.17 | 4.52 | 10.33 | 21.25 |
| ComplEx | 30.84 | 21.51 | 34.48 | 49.58 | 31.69 | 21.44 | 35.74 | 52.04 | 21.01 | 11.87 | 23.47 | 39.87 |
| TTransE | 13.43 | 3.11 | 17.32 | 34.55 | 15.71 | 5.00 | 19.72 | 38.02 | 8.31 | 1.92 | 8.56 | 21.89 |
| TA-Dismult | 26.47 | 17.09 | 30.22 | 45.41 | 24.31 | 14.58 | 27.92 | 44.21 | 16.75 | 8.61 | 18.41 | 33.59 |
| DE-SimplE | 32.67 | 24.43 | 35.69 | 49.11 | 35.02 | 25.91 | 38.99 | 52.75 | 19.30 | 11.53 | 21.86 | 34.80 |
| TNTComplEx | 32.12 | 23.35 | 36.03 | 49.13 | 27.54 | 19.52 | 30.80 | 42.86 | 21.23 | 13.28 | 24.02 | 36.91 |
| RE-Net | 38.28 | 28.68 | 41.34 | 54.52 | 42.97 | 31.26 | 46.85 | 63.47 | 28.81 | 19.05 | 32.44 | 47.51 |
| CyGNet | 32.73 | 23.69 | 36.31 | 50.67 | 34.97 | 25.67 | 39.09 | 52.94 | 24.93 | 15.90 | 28.28 | 42.61 |
| xERTE | 40.79 | 32.70 | 45.67 | 57.30 | 46.62 | 37.84 | 52.31 | 63.92 | 29.31 | 21.03 | 33.51 | 46.48 |
| CENET | - | - | - | - | 37.16 | 27.78 | 41.16 | 55.49 | 27.14 | 18.58 | 29.99 | 44.15 |
| TITer | 41.73 | 32.74 | 46.46 | 58.44 | - | - | - | - | 29.98 | 22.05 | 33.46 | 44.83 |
| RE-GCN | 40.36 | 30.73 | 44.98 | 58.81 | 46.91 | 36.33 | 52.65 | 67.24 | 30.82 | 21.11 | 34.71 | 49.83 |
| DaeMon | - | - | - | - | - | - | - | - | 31.85 | **22.67** | 35.92 | 49.80 |
| TECHS | 43.88 | 34.59 | 49.36 | 61.95 | 48.38 | **38.34** | **54.69** | **68.92** | 30.85 | 21.81 | 35.39 | 49.82 |
| AnyBURL | 29.67 | 21.26 | 33.33 | 46.73 | 32.05 | 23.72 | 35.45 | 50.46 | 22.77 | 15.10 | 25.44 | 38.91 |
| TLogic | 43.04 | 33.56 | 48.27 | 61.23 | 46.97 | 36.21 | 53.13 | 67.43 | 29.82 | 20.54 | 33.95 | 48.53 |
| TR-Rules | 43.32 | 33.96 | 48.55 | 61.17 | 47.64 | 37.06 | 53.80 | 67.57 | 30.41 | 21.10 | 34.58 | 48.92 |
| **INFER** | 44.09 | 34.52 | 48.92 | 62.14 | 48.27 | 37.61 | 54.30 | 68.52 | 31.68 | 21.94 | 35.64 | 50.88 |
| **INFER(with** $vc$**)** | 44.08 | 34.82 | 48.84 | 61.74 | 48.34 | 38.07 | 54.16 | 67.82 | 32.11 | 22.34 | 36.26 | 51.19 |
| **INFER-60(with** $vc$**)** | **44.46** | **35.03** | **49.37** | **62.31** | **48.73** | 38.32 | 54.61 | 68.48 | **32.22** | 22.39 | **36.41** | **51.52** |

Table 1: Performance of INFER on ICEWS14, ICEWS05-15 and ICEWS18. Best results are in bold and second best results are underlined.

**Baseline Methods** We compare INFER with static embedding, temporal embedding and rule-based state-of-the-art models. As for static embedding models we select Dismult (Yang et al., 2014) and ComplEx (Trouillon et al., 2016). Temporal embedding models include TA-DistMult (García-Durán et al., 2018), DE-SimplE (Goel et al., 2020), TNTComplEx (Lacroix et al., 2020), RE-Net (Jin et al., 2019), CyGNet (Zhu et al., 2021a), xERTE (Han et al., 2021), TITer (Sun et al., 2021), RE-GCN (Li et al., 2021), CENET (Xu et al., 2022), DaeMon (Dong et al., 2023) and TECHS (Lin et al., 2023) . The rule-based methods we select are AnyBURL (Meilicke et al., 2019b), TLogic (Liu et al., 2022) and TR-Rules (Li et al., 2023). We reproduce RE-GCN and remove their utilization of static information for fairness and all of the rest results are taken from (Li et al., 2023), (Dong et al., 2023) and (Lin et al., 2023).

## 5.2 MAIN RESULTS

Table 1 displays the results of INFER compared with selected baselines, where INFER(with $vc$) denotes we utilize rules with and without variable constraints to infer instead of only rules without variable constraints and INFER-60(with $vc$) represents we use at most 60 rules for a single query. As we can see, INFER-60(with $vc$) achieves state-of-the-art performance on all datasets with respect to MRR, Hits@1,3,10. Specifically, compared with the best rule-based model TR-Rules, INFER obtains average improvements of 1.35% and 1.55% in MRR and Hits@10 on three datasets, which proves the effectiveness of introducing embedding models and considering the frequency and time validity of historical facts for rule application. Meanwhile INFER maintains the interpretability of rule-based methods. We also provide an exemplary CASE in Appendix C to better illustrate the process of INFER and show the interpretability of INFER. Results of INFER on YAGO and WIKI are presented in Appendix F, Table 5.

| | ICEWS14 | | | | ICEWS18 | | | |
| Model | MRR | Hits@1 | Hits@3 | Hits@10 | MRR | Hits@1 | Hits@3 | Hits@10 |
|---|---|---|---|---|---|---|---|---|
| **INFER** | **44.09** | **34.52** | **48.92** | **62.14** | **31.68** | **21.94** | **35.40** | **50.88** |
| INFER(P+B) | 37.67 | 27.34 | 42.76 | 57.77 | 25.87 | 16.16 | 29.31 | 45.57 |
| INFER(P) | 16.29 | 7.58 | 17.78 | 34.69 | 11.70 | 3.90 | 12.41 | 28.10 |
| INFER(Temp Val) | 41.99 | 33.75 | 47.00 | 57.31 | 30.90 | 21.59 | 35.02 | 49.37 |
| INFER(B) | 37.41 | 27.58 | 42.52 | 56.43 | 25.73 | 16.31 | 29.26 | 44.90 |

Table 2: Ablation study of INFER on ICEWS14, and ICEWS18. Best results are in bold.

## 5.3 ABLATION STUDY

We conduct ablation study on ICEWS14 and ICEWS18 and the results are reported in Table 2 where P denotes the utilization of pre-trained static KG embedding models, B denotes we use the traditional binary truth values for historical facts and Temp Val represents we use the truth values calculated by Temporal Validity Functions.

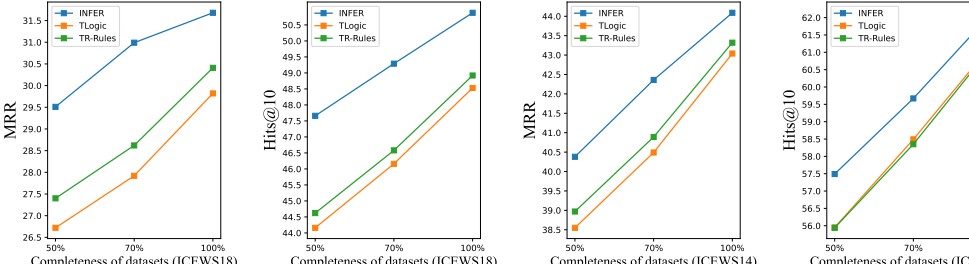

Figure 4: Performance of INFER and other rule-based models on our modified incomplete ICEWS14 and ICEWS18 where a certain proportion of facts at each timestamp are randomly removed.

As we can see from the results of INFER(P+B), when using the binary truth value, the performance of INFER drops dramatically on both datasets with up to 6.42% in MRR and 4.37% in Hits@10. This phenomenon strongly verifies our motivation and proves the necessity of extending truth values to continuous number via modeling temporal validity and frequency. As we do not use the historical information and only project rules based on the matrices given by the static embedding model, the results (INFER(P)) get even worse, which demonstrates the importance of historical facts for rule-based models. INFER(Temp Val) and INFER(B) report the performance when the pre-trained model is removed. Again, compared with the binary settings, the introduction of our Temporal Validity Function significantly improves the performance. The results of INFER(Temp Val) also prove that the introduction of embedding model does contribute to the overall performance, but the the capability of embedding models is not decisive. Besides, by comparing (B) with (P+B) and (Temp Val) with (INFER) itself, we discover that the utilization of Temporal Validity Function amplifies the influence of the pre-trained KG embedding model. We speculate the probable reason is that some of the re-scaled values given by the static embedding model are relatively small compared with **1** when the truth values are binary. Hence, candidates inferred through potential missing facts surely rank after those candidates inferred by existing historical facts. As a result, the effects of the pre-trained model are impaired when binary truth values are utilized.

## 5.4 ANALYSIS

**Performance on Incomplete datatsets**  To demonstrate the capability in inference missing facts and robustness of reasoning on incomplete datasets of INFER, we conduct experiments on modified ICEWS14 and ICEWS18 and compare INFER with previous rule-based methods. We randomly remove 30% and 50% facts at each timestamp and then pre-train the static embedding model and mine temporal rules with these incomplete datasets. Meanwhile, the historical facts used for temporal validity calculation are also cut to 50% and 70%. The results are shown in figure 4. As we can see, the differences between INFER and the other two models generally increase as the datasets become more incomplete. Moreover, the slope of the blue line representing INFER is smaller than that of the other two lines, which indicates that the performance of INFER is less affected as the datasets become sparse. This suggests that INFER exhibits robustness on sparse datasets, which can be attributed to the introduction of the static embedding model that enhances the ability to infer missing facts during rule application, thereby improving the overall performance of INFER.

**Inference Efficiency**  We also test TLogic (Liu et al., 2022) and our method on ICEWS14 to calculate the time consumption, the quantity of processed rules and the number of candidates derived from each rules which serve to evaluate the efficiency and capability of models. In our reproduced experiments, the rule application time of TLogic on ICEWS14 is 2000 seconds with 8 CPUs. The average number of rules TLogic applies for a single query is 32.96 and the number of candidates it covers per rule at each intermediate step is 13.30. In comparison, it takes INFER 2500 seconds to apply rules on ICEWS14 with a single GPU. However, the average number of rules INFER uses for a single query is 38.73 and the average number of candidates per rule that INFER considers and assign positive scores to at each step is 1385.49.

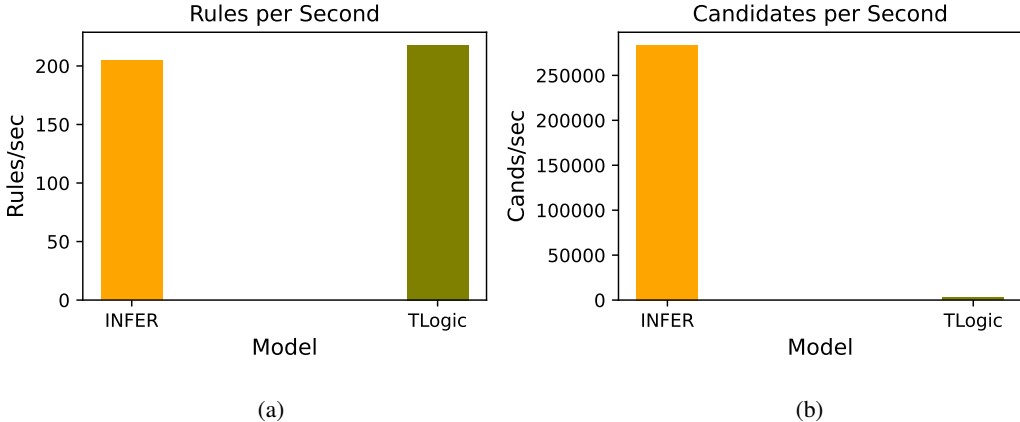

Figure 5: Comparison on inference efficiency: (a) Comparison on rules/sec, INFER gives similar performance to TLogic. (b) Comparison on candidates/sec, INFER shows significantly superiority which means that INFER achieve efficient traversal on graphs.

In contrast, INFER can apply rules more efficiently under our proposed paradigm and significantly enlarge the range of candidate entities which surely contributes to the final performance of INFER since more considered intermediate candidates denote that our model traverse more possible paths during application.

**Variance**  Generally, INFER can be viewed as a statistical learning method. Thus, as long as the mined rules and corresponding confidence are the same, when running the model with the same hyper-parameter configuration, it will yield identical results. The only possible variance comes from the pre-trained static KG embedding model. To eliminate this concern, we pretrain the ComplEx 3 times on ICEWS14 and subsequently run INFER. The results of the three experiments are completely identical. The possible reason is that the scores given by the KG embedding model are normalized by a softmax function and filtered with a threshold, which effectively avoid any obvious fluctuation. Thus, we can conclude that the performance of INFER is stable and the gained improvements are not occasional issues.

**Variable Constraints**  As we can see in table 1, when the limit of the quantity of rules is set to 40, the utilization of rules with variable constraints does not bring significant improvements. We speculate that rules with variable constraints require more strict temporal restriction which can not be satisfied by current design. Meanwhile, the inference speed of applying rules with variable constraints decreases due to extra indexing and calculations. Detailed process of applying rules with variable constraints is shown in Appendix A.

## 6 CONCLUSION

In this paper, we propose INFER, an interpretable neural-symbolic model for TKG extrapolation. INFER proposes a novel paradigm for integrating temporal rules with embedding-based methods which enhances the robustness of rule-based methods to missing facts. INFER is the first model to extend the truth value of facts into continuous number and designs Temporal Validity Function to model the time validity and frequency of historical facts. We introduce a rule projection module which manages to apply rules by conducting faster matrices operations on GPUs instead of previous costly path matching in graphs. Experimental results show that INFER achieves state-of-the-art performance on three TKG datasets. Meanwhile, INFER also remarkably improves the efficiency of rule application.

**Reproducibility Statement.** We provide our code at `https://github.com/JasonLee-22/INFER` to enhance reproducibility. And the implementation details are provided in section 5.1 and Appendix E.

ACKNOWLEDGEMENTS

This work was supported by the National Natural Science Foundation of China (No. 62473271) and ( No. 62176026).

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

## A    RULE PROJECTION

### A.1    VARIABLE CONSTRAINTS

As we mentioned in section 3.2, in some rules, entity variables in the rule bodies do not have to be unique since the same entity might be involved in multiple facts at different timestamps. This phenomenon which is called Variable Constraints brings constraints to the rule application procedure. Here are some examples:

$$(A, support, C, T) \because (A, riot, B, T_1) \bigwedge (B, make\ statement, A, T_2) \bigwedge (A, resort, C, T_3)$$

The above example is a rule with variable constraints. As we can see in the body of the rule, in the second hop, the entity which $B$ make statement to has to be the same with the entity that riot with $B$ at $T_1$.

However, the below case is a rule without variable constraints which is different with the rule given above. Because the entity which $B$ make statement to can be or can not be the same with the entity riot with $B$ at $T_1$.

$$(A, support, C, T) \because (A, riot, B, T_1) \bigwedge (B, make\ statement, D, T_2) \bigwedge (D, resort, C, T_3)$$

## A.2 RULE PROJECTION WITH VARIABLE CONSTRAINTS

Algorithms 2 and 3 present the pseudocode for rule projection with two types of variable constraints: [1,3] and [0,2], assuming a rule length of 3, where the numbers in the list indicate that the corresponding entity variables in the rule body must refer to the same entity. In our experiments, we also mine rules with variable constraints [[1,3],[0,2]] and rules of length 2 from ICEWS0515 with variable constraints [0,1] and [1,2]. Here, we provide the projection algorithms for the two most common types of variable constraints. Algorithms for these 3 variable constraints can be easily derived following Algorithms 2 and 3 and our provided code can serve as a reference.

---

**Algorithm 2** Pseudocode of Rule Projection With Variable Constraints [1,3]

---

**Require:** $(s, r_q, ?, T_q)$: Query; $\{R_1, R_2...R_n\}$: Corresponding rule bodies; $M_\mathcal{R}$: Temporal Weight Matrices;

1: set $\mathbf{Ans} = \mathbf{M_{R_1}}[s,:]$    # Select the s-th row of $\mathbf{M_{R_1}}$
2: $\mathbf{Ans} = \mathbf{Ans^T}.\textbf{repeat}(1, |\mathcal{E}|)$    #Repeat the transposed vector along row direction
3: $\mathbf{Ans} = \mathbf{Ans} \odot \mathbf{M_{R_i}}$    #Hadamard Product gives a matrix of $\mathbf{R}^{\mathcal{E}*\mathcal{E}}$, where $\mathbf{Ans}[i][j]$ denotes the score of entity $e_j$ when entity $e_i$ serves as the constrained variable.
4: $\mathbf{res} = \text{zeros}(1, |\mathcal{E}|)$
5: **for** $i$ **in** $(0, |\mathcal{E}|)$ **do**
6:     row = $\mathbf{Ans}[i,:]$
7:     row = $\text{row}^\mathbf{T}.\textbf{repeat}(1, |\mathcal{E}|)$
8:     $\mathbf{temp} = \text{row} \odot \mathbf{M_{R_3}}$
9:     $\mathbf{temp} = \mathbf{temp}[:, i]$
10:     res[i] = $\textbf{max}(\mathbf{temp}, dim = -1)$
        **return res**

---

**Algorithm 3** Pseudocode of Rule Projection With Variable Constraints [0,2]

---

**Require:** $(s, r_q, ?, T_q)$: Query; $\{R_1, R_2, R_3\}$: Corresponding rule bodies; $M_\mathcal{R}$: Temporal Weight Matrices;

1: set $\mathbf{Ans} = \mathbf{M_{R_1}}[s,:]$    # Select the s-th row of $\mathbf{M_{R_1}}$
2: $\mathbf{Ans} = \mathbf{Ans^T}.\textbf{repeat}(1, |\mathcal{E}|)$    #Repeat the transposed vector along row direction
3: $\mathbf{Ans} = \mathbf{Ans} \odot \mathbf{M_{R_2}}$    #Hadamard Product
4: $\mathbf{Ans} = \mathbf{Ans}[:, s]$    #Only select the s-th column to satisfy the variable constraints.
5: $\mathbf{Ans} = \mathbf{Ans}.\textbf{repeat}(1, |\mathcal{E}|)$
6: $\mathbf{Ans} = \mathbf{Ans} \odot \mathbf{M_{R_3}}$    #Hadamard Product
7: $\mathbf{Ans} = \textbf{max}(\mathbf{Ans}, dim = -1)$    #Column-wise max operation giving a row vector
        **return Ans**

---

In our preliminary conception, variable constraints of longer rule are solvable without having to traverse every possible scenario for algorithm design. In this conception, we can treat the algorithms for variable constraints that have been designed for lengths of 2 or 3 as algorithm units. The calculations for variable constraints in longer rules can be broken down into combinations of these units. For example, for a rule of length 5 with variable constraints [[0,2,4]]:

$$(A, R_1, B) \wedge (B, R_2, A) \wedge (A, R_3, C) \wedge (C, R_4, A) \wedge (A, R_5, D)$$

We can first use existing algorithms to compute the results under the [0,2] constraint when reaching variable#2 (A). This intermediate result can then serve as the initial condition for the third relation (R3) calculation in the rule. At this point, the subsequent calculation can be seen as a computation for a rule of length 3 with variable constraints [0,2] ([2,4] → [0,2]) : $(A, R_3, C) \wedge (C, R_4, A) \wedge (A, R_5, D)$. This is again a problem that can be solved by our algorithm units.

This approach allows us to tackle longer rule variable constraints by leveraging the solutions for shorter ones, effectively avoiding exhaustive searches through all possible scenarios. By decomposing complex problems into simpler, already-solved components, we can design more efficient and scalable algorithms for handling variable constraints in various rule lengths. We leave the detailed design of a more flexible and automatic algorithm for handling variable constraints in long rules to the future work.

# B  STATISTICS OF DATASETS

Table 3 shows the statistics of these five datasets. To satisfy the extrapolation reasoning settings, facts are sorted in ascending order based on timestamps and then split into train, valid and test.

|  | $Train$ | $Valid$ | $Test$ | $|\mathcal{E}|$ | $|\mathcal{R}|$ | $|\mathcal{T}|$ |
|---|---|---|---|---|---|---|
| ICEWS14 | 63685 | 13823 | 13222 | 7128 | 230 | 365 |
| ICEWS0515 | 322958 | 69224 | 69147 | 10488 | 251 | 4017 |
| ICEWS18 | 373018 | 45995 | 49545 | 23033 | 256 | 304 |
| WIKI | 539286 | 67538 | 63110 | 12554 | 24 | 232 |
| YAGO | 161540 | 19523 | 20026 | 10623 | 10 | 189 |

Table 3: Statistics of ICEWS14, ICEWS05-15, ICEWS18, WIKI and YAGO.

# C  EXEMPLARY INFERENCE CASE

Figure 6 displays an inference case of INFER. Given the query (AshrafGh ani Ahmadzai, Make a Visit, ?, 2014/11/11) whose answer is China and corresponding rule of length 3, we list top 5 candidates at each step of rule application provided by INFER. As for the result, INFER successfully ranks China at the second place. Moreover, we discover that the other 3 entities: Afghanistan, Jens Stoltenberg and North Atlantic Treaty Organization also serve as the object entity for this query although at other timestamps, which demonstrates that INFER is indeed capable of obtaining relevant candidates.

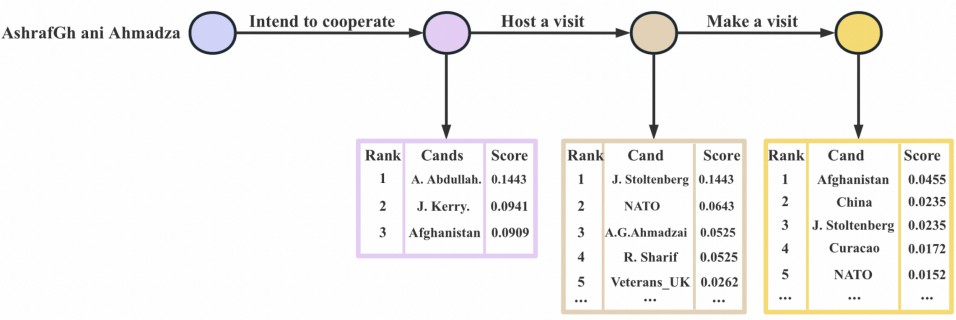

Figure 6: Exemplary interpretable inference procedure of INFER.

# D  TEMPORAL VALIDITY FUNCTION

Table 4 reports the performance of various forms of Temporal Validity Function on ICEWS14. We mainly focus on the main time span part, where Square root, Cube root, Reci and Exp represent the functions $\frac{1}{\sqrt{x}}$, $\frac{1}{\sqrt[3]{x}}$, $\frac{1}{x}$ and $\frac{1}{exp(x)}$ used for the time span part respectively and Binary denotes the previous binary truth value settings for historical facts. As we can see, following our intuition, by proposing a function which is negative to the time span values and ranging in [0,1], INFER can properly model the temporal validity of historical facts. However, Exp gives worse performance compared with Binary, it is natural to speculate that it is that $\frac{1}{exp(x)}$ drops too fast as the time span getting bigger leading to the "forget" of historical facts. From this point of view, when using the Binary settings, the temporal validity of facts do not decay, i.e. the model strongly memorize every fact. Thus, Binary and Exp are two extreme cases, we try to explore a more suitable form to model the temporal validity and select the Square root in INFER, which is still a primary attempt and we believe more sophisticated Temporal Validity Function can surely improve the performance of INFER.

| Model | ICEWS14 | | | |
|---|---|---|---|---|
| | MRR | Hits@1 | Hits@3 | Hits@10 |
| **INFER** | **44.09** | **34.52** | **48.92** | **62.14** |
| Square root | 44.09 | 34.52 | 48.92 | 62.14 |
| Cube root | 43.76 | 34.18 | 48.54 | 62.01 |
| Reci | 41.36 | 31.52 | 45.94 | 60.61 |
| Exp | 33.08 | 23.58 | 36.87 | 52.18 |
| Binary | 37.67 | 27.34 | 42.76 | 57.77 |

Table 4: Results of various forms of Temporal Valid Function in INFER on ICEWS14. Best results are in bold.

# E  IMPLEMENTATION DETAILS

When mining rules, we use the configuration given by TR-RulesLi et al. (2023). We select ComplEXTrouillon et al. (2016) as the static KG embedding model, which is trained with N3 regularizor and relation prediction task. $\tau$ in equation (2) is set 15 for ICEWS14 and ICEWS18. While for ICEWS0515 which covers a time span of ten years, we set it to 150. For some queries, there might be too many rules can be utilized for inference. However, for simplicity and resource saving, we set a limitation on the most rules can be used for a single query which is 40 unless specified and the threshold for building temporal weight matrices is set to 0.0005.

# F  RESULTS ON YAGO AND WIKI

Table 5 reports the results of INFER on WIKI and YAGO, as we can see INFER still outperforms existing methods and significantly improves the rule-based baselines on WIKI. As for YAGO, our model and other rule-based methods fall short compared with embedding-based methods. As mentioned in (Huang et al., 2024), some relations in YAGO can not be modeled by cyclic rules which take up to 10% of the test set. Since our model only uses cyclic rules, it is affected to a certain extent. However, INFER still gain improvements compared with TLogic. It should be noted that during experiments, we notice that the introduction of "generated rule" mentioned in section 4.4 causes bias on the YAGO dataset, which drastically boosts the performance. Thus, we do not leverage the "generated rules" for evaluation on YAGO for fair comparison. We speculate the possible reason is that there are more repetitive facts in YAGO which amplifies the impact of the "generated rules" too much especially when given the fact that less patterns can be modeled with cyclic rules. We also conduct experiments on the other four datasets to investigate the bias issue and the results show that it only yields slight fluctuation to the overall performance on the other four datasets.

| Model | WIKI | | | | YAGO | | | |
|---|---|---|---|---|---|---|---|---|
| | MRR | Hits@1 | Hits@3 | Hits@10 | MRR | Hits@1 | Hits@3 | Hits@10 |
| **INFER** | **86.48** | **85.09** | **87.38** | **88.67** | 83.74 | **83.54** | **83.72** | 84.31 |
| TLogic | 78.93 | 73.05 | 84.97 | 86.91 | 78.76 | 74.31 | 83.38 | 83.72 |
| TECHS | 75.98 | - | - | 82.39 | **89.24** | - | - | **92.39** |
| TITER | 73.91 | 71.70 | 75.41 | 76.96 | 87.47 | 80.09 | 89.96 | 90.27 |
| REGCN | 78.53 | 74.50 | 81.59 | 84.70 | 82.30 | 78.83 | 84.27 | 88.58 |

Table 5: Results of INFER on WIKI, and YAGO. Best results are in bold.

| Model | ICEWS14 | | | |
|---|---|---|---|---|
| | MRR | Hits@1 | Hits@3 | Hits@10 |
| INFER-1 | **43.83** | **34.15** | **49.01** | **61.99** |
| TR-Rules-1 | 40.59 | 30.90 | 46.31 | 59.14 |
| INFER-2 | **20.49** | **10.61** | **22.97** | **42.38** |
| TR-Rules-2 | 15.28 | 6.40 | 16.82 | 35.29 |
| INFER-3 | **40.69** | **32.48** | 45.40 | 55.99 |
| TR-Rules-3 | 40.45 | 31.46 | **45.52** | **57.35** |

Table 6: Results of INFER with rules of different lengths on ICEWS14. Best results are in bold. (Model-X, X denotes only use rules of length X.)

## G  LIMITATIONS

Although INFER obtains significant improvements on three datasets, there are still limitations. As we mentioned in section 6, the main limitation of INFER is that it can not explicitly model the exact order of historical facts which might be crucial for improving the performance of longer rules with variable constraints. We provide the performance of INFER using rules of different lengths with comparison to TR-Rules in Table 6. It can be observed that, INFER greatly improves the effect of applying rules of length of 1 and 2 compared to TR-Rules. As for length of 3, INFER still performs better on MRR and Hits@1 but the overall improvements are not that significant compared with shorter rules. Besides, INFER can not leverage acyclic rules for inference, which have been demonstrated crucial in KG completion.

