# OpenReview forum: "INFER: A Neural-symbolic Model For Extrapolation Reasoning on Temporal Knowledge Graph"
_ICLR.cc/2025/Conference — ICLR 2025 Poster_

### Official Review · Reviewer_qjZA · 2024-10-31

**Soundness:** 3
**Presentation:** 2
**Contribution:** 2
**Rating:** 5
**Confidence:** 4

**Summary:**

This paper introduces INFER, a neural-symbolic model designed for Temporal Knowledge Graph (TKG) extrapolation reasoning. Traditional rule-based methods for TKGs, though interpretable, struggle with temporal reasoning as they treat facts as binary and ignore temporal frequency and validity. NFER addresses these issues through a Temporal Validity Function, which enables continuous truth values and models the frequency and validity of historical facts for better temporal adaptation. Additionally, INFER incorporates Temporal Weight Matrices with a pre-trained static KG embedding model to improve inference quality. A rule projection module enhances computational efficiency by leveraging GPU-optimized matrix operations, allowing INFER to scale effectively and integrate with embedding-based approaches. Experiments show that INFER achieves state-of-the-art results on several datasets, demonstrating enhanced inference capabilities over existing models, particularly in sparse TKG settings.

**Strengths:**

1.   Temporal Knowledge Graph (TKG) reasoning is an important  research topic.
 2.   The proposed method shows better results compared to existing rule-based methods.

**Weaknesses:**

1.  The motivation behind the method design is unclear, and the description lacks clarity. For example, in Section 4.3.3, the rationale for the rule projection strategy is not well-explained, and the meanings of terms like “Ans_i” and “Ans” are not clarified.
2.  Limited novelty. The techniques used in the proposed method do not introduce significant innovations.
3.  The temporal and spatial complexity of the inference process appears high, which might impact practical applicability. It would be beneficial to provide an analysis of the computational complexity.
4.  Some experimental details are insufficiently explained. For example, in Line 426, the phrase “traditional binary truth values for historical facts” needs further clarification. It is unclear what specific criteria or methods are used to assign these binary truth values to historical facts.
5.  The dataset coverage is insufficient. The paper only uses the ICEWS dataset, which belongs to a specific category of TKG data. It would be valuable to include additional datasets like WIKI or GDELT to demonstrate the method’s generalizability across different data types.

**Questions:**

See Weaknesses.

---

> ### Author Response · Authors · 2024-11-16
> **Reply to Reviewer qjZA**
>
> Thank you for your insightful feedbacks on our paper. We appreciate your crucial suggestions. The followings are our replies to your questions and summarized weaknesses:
>
> Reply to W#1:
>
> We have two main motivations. Firstly, traditional rule-based models, when applying rules through path matching, do not consider the validity of historical facts and treat recent facts and ancient facts equally (lines 82-84 in our paper). Secondly, traditional rule-based models are greatly affected by missing information, leading to inference bottlenecks. To address the first issue, we propose a temporal validity function that considers the time span and frequency of historical facts, extending the truth value of facts from binary to continuous and modeling fact validity. The main experiment and ablation experiment verify this motivation. To address the second issue, we introduce a static KG embedding model, which captures structural information through pre-training. We then propose the entire INFER paradigm, which combines embedding information with improved rule application by constructing and updating the Temporal Weight Matrix and proposing Projection operator. The resulting neural-symbolic method gains the ability to reason about missing information and its performance is less affected under incomplete settings compared to traditional rule-based models (Results in Table 2). Meanwhile, the efficiency of applying rules is greatly enhanced through performing matrices operations instead of path matching. More importantly, our model provides a new paradigm to combine embedding-based methods and symbolic-based methods paving the way for future research.
>
> As we mentioned in section 4.3.3, when answering query $(s,r,?,t)$, we first traverse the available rule’s body obtaining the relations $rl_1,rl_2$.... At the starting point we select the $s$-th row of $M_{rl_1}$ which records the truth values of each entity serving as object entity in facts with subject entity $s$ and relation $rl_1$, $(s,rl_1,e_i);e_i∈E$. If the rule length is 1, then this vector denotes the scores of all entities given by this rule. If the rule length is more than 1, as mentioned in the paper we transpose the row vector and repeat it along the row direction for $|E|$ times which results in a matrix of $|E|×|E|$. Then we calculate the Hadamard product of the obtained matrix and the corresponding temporal weights matrix, which is the results of the **AND** of facts along each possible paths. Finally, we take the maximal value of each column, which is to select the biggest truth value of a path starting from s and ending at each entity $e_i$. This process is tractable and can be interpreted as human-readable results. For example by performing argmax operation, we can show which specific path leads to the maximal score of $e_i$. As we repeat the above procedure, the final results is the vector records the maximal truth values of paths leading to each entity.
>
> We are sorry about the confusing presentation. In equation 5, **$Ans_i$** is a row vector which denotes the scores of all candidates at the $i$-th hop of a rule and n is the length of the rule. As for the Ans in Algorithm 1, it is also a row vector representing the scores of all candidates at each step.
>
> Reply to W#3:
>
> Thanks for your advice. Theoretically, the spatial complexity of INFER mainly counts on the temporal weight matrices: O(|E| * |E| *|R|). Although it seems costly, TKGs in real world are really sparse and we can use sparse matrix to store them. Also, as mentioned in line 259, we set a threshold to filter out some really small values so that the matrices can be stored on a single GPU. Here’s some statistics to illustrate the sparsity of TKGs:
>
> For ICEWS14, the facts it contains take up only 0.00043% of the whole matrices, which means the rest entries in the Temporal Weight Matrices are zero. The proportion of ICEWS18 is 0.00020%, 0.00060% for ICEWS0515.
>
> As for the time complexity, if the length of the rule is 1, then we just select a specific row of the corresponding Temporal Weight Matrix $M_r$. Thus, the time complexity is $O(1)$.
>
> Intuitively, when the rule is longer than 1, the time complexity of the inference procedure of INFER is $O(|E|^2)$ due to the Hadamard Product, which seems high. However, due to the sparse property of TKGs, in fact there are many zero values in $Ans$ (the vector stores intermediate candidates list), thus we actually implement it in a more efficient way through only selecting the non-zero values in Ans and calculate the product of the values and their corresponding rows in $M_r$. In this way, the time complexity is $O(|E| * len(Ans>0))$.

---

> ### Author Response · Authors · 2024-11-16
> **Reply to Reviewer qjZA #2**
>
> Reply to W#4:
>
> As we mentioned in Section 1, line(82-84), “previous rule-based methods consider the standing of facts as a binary 0 or 1 problem (have been true or not) for path matching, which ignores the validity as well as frequency of historical facts under temporal settings”. The reason why we have the above claim is that when applying rules, previous methods search for the stands of rule bodies through paths matching on the graphs. If we consider the knowledge graphs as weighted graphs, then normally there are only two kinds of weights: 0 and 1. 0 for the facts that have never appeared, i.e. there’s no edges between two entity nodes. 1 for the facts that have been true in history, i.e. there’s an edge of a specific relation between two entity nodes. In this paper, we argue that this setting is not suitable for TKGs. Details can be found in line 97-94 and 107-109.
>
> So back to your question, “traditional binary truth values for historical facts” in line 426 means that we assign 1 to all historical facts as long as it appeared before, otherwise we do not update its value in the temporal weight matrices.
>
> Reply to W#5:
>
> Here are the results of INFER on WIKI and YAGO, as we can see INFER still outperforms existing  methods and significantly improves the rule-based baselines on WIKI. As for YAGO, our model and other rule-based methods fall short compared with embedding-based methods. As mentioned in [1], some relations in YAGO can not be modeled by cyclic rules which take up to 10% of the test set. Since our model only uses cyclic rules, it is affected to a certain extent. However, INFER still gain improvements compared with TLogic.
>
> **WIKI:**
> | Model     | MRR     | Hits@1 | Hits@3 | Hits@10 |
> | -------- | -------- | -------- | -------- | -------- |
> | REGCN | 78.53 |  74.50 | 81.59 | 84.70 |
> | TITER | 73.91 |  71.70 | 75.41 | 76.96 |
> | TECHS | 75.98 |  - | - | 82.39 |
> | TLogic | 78.93 |  73.05 | 84.97 | 86.91 |
> | INFER | **86.48** | **85.09** | **87.38** | **88.67** |
>
>
> **YAGO:**
> | Model     | MRR     | Hits@1 | Hits@3 | Hits@10 |
> | -------- | -------- | -------- | -------- | -------- |
> | REGCN | 82.30 |  78.83 | 84.27 | 88.58 |
> | TITER | 87.47 |  80.09 | 89.96 | 90.27 |
> | TECHS | **89.24** |  - | - | **92.39** |
> | TLogic | 78.76 |  74.31 | 83.38 | 83.72 |
> | INFER | 83.74 | 83.54 | 83.72 | 84.31 |
>
> [1] Confidence is not Timeless: Modeling Temporal Validity for Rule-based Temporal Knowledge Graph Forecasting (Huang et al., ACL 2024)
>
> Please feel free to reach out if you have any questions or require further clarification. If you find our response satisfactory, we hope you will consider this a valid reason to consider raising your rating.

---

> ### Author Response · Authors · 2024-11-24
> **Looking Forward to Your Reply**
>
> Dear Reviewer qjZA,
>
> I hope this message finds you well. As the deadline for the reviewer-author discussion draws near, we would like to kindly request your input on our rebuttal. We understand your time is valuable, and we greatly appreciate your initial engagement with our paper.   If you could spare a moment to review our responses and consider the clarification and extra experiments we've made, we would be truly grateful.
>
> Thank you once again for your time. Please feel free to reach out if you have any questions or require further clarification. If you find our response satisfactory, we hope you will consider this a valid reason to consider raising your rating.

---

> > ### Comment · Reviewer_qjZA · 2024-11-26
> >
> > Thank you for your response and clarification. I will maintain my score.

---

### Official Review · Reviewer_EcyD · 2024-11-01

**Soundness:** 3
**Presentation:** 2
**Contribution:** 3
**Rating:** 5
**Confidence:** 4

**Summary:**

To solve the problem that rule-based methods in the field of temporal knowledge graph inference have insufficient reasoning ability when graph facts are missing, and the fact state is simply considered as binary, ignoring the validity and frequency of historical facts, this paper proposes the neural symbol model of INFER, which quantifies fact credibility in time dimension by introducing a time validity function. At the same time, the time weight matrix is introduced so that the model can infer the missing facts and deal with the incompleteness of the map. And to improve the efficiency of rule reasoning, a rule projection module is proposed, which uses GPU-based matrix operation instead of traditional path matching.

**Strengths:**

(1) This paper introduces effective means such as the time validity function to quantify fact credibility in the time dimension and proves its effectiveness through experiments, which provides ideas for how to improve the reasoning ability of rule-based methods in dealing with incomplete graphs.
(2) The average performance of the experiment is good on multiple datasets, and the self-made TKG data with sparse facts is obtained. The experimental results show that the INFER model is significantly better than other methods when the facts are sparse.
(3) The chart is clear, the paper is completed with a high degree, and it is easy to interpret.

**Weaknesses:**

(1) The Design of the time validity function: The time validity function proposed in this paper calculates the time weight of historical facts based on the time interval and frequency of fact occurrence. Although the above two terms are considered at the same time, the function form is relatively fixed and more dependent on experience. Adapting the attenuation rate using data-driven methods may enhance the model's adaptability.
(2) The performance of the model on the ICEWS05-15 dataset does not exceed that of TECHS, and there is no detailed analysis of the results in this paper.
(3) The rule projection module used by INFER loses the ability to directly model the sequence of facts to a certain extent, which may affect the accuracy in scenarios requiring strict time order or multi-jump reasoning. Additional experiments are needed for evaluation, especially for long rule samples with variable constraints.
(4) INFER introduces neural network embedding and complex matrix operations, and although the authors show entity scores when the rules are applied, it still damages the interpretability of the traditional rule model to some extent.

**Questions:**

(1) Is the use of the square root form of the time decay term in the time validity function based on the conclusions obtained from experiments?
(2) What is the additional overhead if you replace the static embedded model with a time-embedded model? And What's the performance boost?
(3) When calculating the rule confidence, the INFER model adopts the rule learning algorithm in TRRules. The rule base retains only cyclic rules and filters out acyclic rules. TR-Rules uses acyclic rules and proves its effectiveness. What is the reason for filtering acyclic rules here? Is there any experimental support?
(4) Does the static embedding model provide the same level of confidence for the same fact at different time points?
(5) How does INFER's rule projection module perform when dealing with rules of different lengths?

---

> ### Author Response · Authors · 2024-11-16
> **Reply to Reviewer EcyD**
>
> Thank you for your feedbacks and here are our clarifications about your questions:
>
> Reply to W#1:
>
> The temporal validity function we proposed is indeed an empirical result. When designing the model, we also considered that using a learnable approach might achieve better performance. However, our current framework is already divided into three stages: rule learning, pre-trained model training, and rule application. Introducing an additional learning and training process for the temporal validity function would make the overall workflow cumbersome compared to previous rule-based methods. Through experiments, we found that designing a fixed temporal validity function can also achieve good results. Therefore, we adopted the current approach. However, we fully agree with your viewpoint that a learnable approach may provide better performance, and we will attempt to explore this issue in future work.
>
> Reply to W#2:
>
> Since our model uses an existing algorithm (TR-Rules) to acquire rules during the rule learning stage, the performance of INFER is dependent on the quality of the learned rules. As can be seen from Table 1, compared to ICEWS14 and ICEWS18, the performance of TR-Rules on ICEWS0515 is the furthest from that of TECHS. This is because ICEWS0515 contains data spanning 10 years and is relatively sparse, which increases the difficulty of capturing stronger causal rules and assessing rule confidence during the rule learning process. Therefore, the relatively low quality of the learned rules affects the final performance of INFER. However, we can see that on ICEWS0515, INFER has significant improvements in MRR and Hits@1 compared to TR-Rules, which demonstrates the effectiveness of our proposed rule application paradigm.
>
> As mentioned in Appendix F, INFER's current inability to effectively model the sequential order of events may, to a certain extent, limit the its current capabilities. We will also explore effective ways to address this issue in our future work. Below is a comparison of INFER's performance when applying rules of different lengths with TR-Rules on the ICEWS14 dataset:
>
> **Length 1:**
> | Model     | MRR     | Hits@1 | Hits@3 | Hits@10 |
> | -------- | -------- | -------- | -------- | -------- |
> | TR-Rules | 40.59 |  30.90 | 46.31 | 59.14 |
> | INFER | **43.83** | **34.15** | **49.01** | **61.99** |
>
> **Length 2:**
> | Model     | MRR     | Hits@1 | Hits@3 | Hits@10 |
> | -------- | -------- | -------- | -------- | -------- |
> | TR-Rules | 15.28 |  6.40 | 16.82 | 35.29 |
> | INFER | **20.49** | **10.61** | **22.97** | **42.38** |
>
> **Length 3:**
> | Model     | MRR     | Hits@1 | Hits@3 | Hits@10 |
> | -------- | -------- | -------- | -------- | -------- |
> | TR-Rules | 40.45 |  31.46 | **45.52** | **57.35** |
> | INFER | **40.69** | **32.48** | 45.40 | 55.99 |
>
> It can be observed that, INFER greatly improves the effect of applying rules of  length of 1 and 2 compared to TR-Rules. (Rules with length of 2 also include variable constraints sometimes) As for length of 3, INFER still performs better on MRR and Hits@1. However, the results of INFER have more bias when dealing with rules of length 2 or 3. Because, we use 40 rules during evaluation and there are less high confidence (quality) rules of length 2 or 3 compared with rules of length 1, which means when evaluating rules of length 3, we might use some low quality rules which further lead to undesirable  results.
>
> Reply to W#4:
>
> We fully understand your concerns. We believe that, compared to traditional rule-based methods, the current interpretable process presented by INFER can provide scores for candidates, only lacks the path that leads to each candidate at each step. However, this issue can be resolved. We only need to add a step of argmax operation before line 5 in Algorithm 1, so that we can obtain the specific path indicating which entity from the previous hop leads to each candidate with the highest score. After obtaining such results, the reasoning process we provide will be completely consistent with traditional rule-based methods.
>
> Reply to Q#1:
>
> Yes, as we mentioned in line(295,296), it is an empirical results. We tested multiple designs of this function and corresponding results are reported in Appendix D, Table 4.

---

> ### Author Response · Authors · 2024-11-16
> **Reply to Reviewer EcyD #2**
>
> Reply to Q#2:
>
> Currently we have $|R|$ temporal weight matrices and the size of each of them is $|E| * |E|$. If we use a temporal embedding model, then for $|T|$ timestamps, we need to build $|R|$ matrices of $|E| *|E|$ for each of them, since each fact might have a different score at different timestamp. This is the overhead issue.
>
> However we choose to use static embedding model not only because of the overhead issue:
>
> Temporal embedding models can only score facts at timestamps that they have seen during the training stage. However, under extrapolation settings, the timestamps in the training set do not overlap with the timestamps in the validation and test set.  When we use the matched bodies appeared after the training set timestamps but before the query timestamps, the temporal model can not give a reasonable score to them. Nevertheless, static embedding models can learn the overall structural information and give fair scores based on historical facts for future usage.
>
> Secondly, as we mentioned in line 256-258, during rule application, unlike traditional rule-based models’ path matching way, INFER does not consider the specific timestamp of facts. Thus, we do not need a score of a facts at a specific timestamp. The structural information of the whole historical facts better serves as the dependence of probabilities of potential missing facts.
>
> Reply to Q#3:
>
> We have observed that the improvement brought by acyclic rules in TR-Rules is not very significant, and the relatively large number of such acyclic rules may result in greater overhead. (TR-Rules report that the number of acyclic rules used is far greater than the number of cyclic rules) Therefore, we have not designed to utilize acyclic rules so far. As mentioned in Appendix F, we will explore this aspect in our future work.
>
> Reply to Q#4:
>
> Yes. Since the pre-trained static embedding model scores each fact ignoring the timestamps, it gives same score to the same fact at different timestamps.
>
> Reply to Q#5:
>
> Please see the results given in the reply#1 to W#3.
>
> If you still have any concerns, please feel free to comment.  If you find our response satisfactory, we hope you will consider this a valid reason to consider raising your rating.

---

> ### Comment · Reviewer_EcyD · 2024-11-21
>
> Thank you for getting back to me. My concerns have been largely addressed, and I will raise my score

---

### Official Review · Reviewer_cyuQ · 2024-11-02

**Soundness:** 3
**Presentation:** 3
**Contribution:** 3
**Rating:** 6
**Confidence:** 4

**Summary:**

The article presents a detailed neuro-symbolic model addressing the task of Temporal Knowledge Graph Completion, particularly focusing on the challenge of extrapolation. This task aims to infer knowledge for a given Knowledge Graph at time T using only past data. The authors propose a well-structured solution, providing an extensive evaluation that compares their model to current state-of-the-art symbolic and neural methods.

**Strengths:**

1. Comprehensive Evaluation: The paper includes a thorough evaluation of the proposed model, supplemented by an ablation study and an efficiency analysis, making it easy to understand the model's effectiveness.
2. Efficient Model Design: The model circumvents the common complexity issue of creating separate matrices per timestamp, resulting in a compact, efficient design.
3. Clear and Reproducible Description: The methodology is well-articulated, allowing for straightforward reproduction of the results.

**Weaknesses:**

1. Minor Typos and Inconsistencies:
   - Line 451/452: The term "INFER(Temp)" is used instead of "INFER(Temp Val)."
   - Line 515: The rule quantity is stated as 40, while Table 1 lists it as 60, which could lead to some confusion.
2. Efficiency Study Observations: The conclusions drawn in the efficiency study are somewhat unclear. Specifically, a competing model achieving a similar score while exploring fewer candidates might suggest that the alternative approach is better optimized in its candidate selection process. Clarifying these efficiency aspects would strengthen the study.
3. Completeness of Graph Argument: The argument concerning graph completeness and the slope behavior appears overstated, as the trend lines for the three methods are quite similar. Revising this interpretation could enhance clarity.
4. Ambiguity in Section 4.3.3: The fourth paragraph lacks clarity regarding "variable constraints in rules," which limits the reader's understanding of the approach. A clearer explanation or example would be beneficial here.

**Questions:**

1. In Section 4.3.2, the function $V(s, r, o, t_c)$ is introduced. However, it is unclear how the model handles $t_{last}$ if the fact $<s, r, o>$ was never previously observed. Is the value set to 0, potentially conflicting with timestamp 0, or is another value assigned?
2. Could the authors elaborate on the benefits of examining a significantly higher number of candidates (100x), especially given the notable increase in runtime (an additional 500 seconds)? Understanding the trade-offs would be helpful.

---

> ### Author Response · Authors · 2024-11-16
> **Reply to Reviewer cyuQ**
>
> Thank you for your insightful feedbacks on our paper. We appreciate your crucial suggestions. The followings are our replies to your questions and summarized weaknesses:
>
> Reply to W#1:
>
> Thank you for pointing out the issues, and we will make the necessary modifications in the revised PDF. Regarding the number of rules, we provided an explanation for "60" on lines 406-408 of our paper, indicating that it represents the performance of INFER when using 60 rules for testing.
>
> Reply to W#2:
>
> Thank you very much for your constructive suggestions, which have helped us improve the expression of our paper. The reason why TLogic uses fewer candidates is due to computational cost constraints, as TLogic sets an **upper limit** of **20** candidates to consider. Once the number of searched candidates reaches 20, the TLogic algorithm will directly stop applying more rules to obtain more candidates, meaning that it does not consider the potential impact of additional rules on the current results. Therefore, it does not provide a superior method for selecting candidates but rather exhibits limited performance due to cost considerations. Our proposed INFER can efficiently consider a wider range of candidates to obtain more accurate results. Thus, we believe that INFER has higher efficiency, which allows it to consider more rules and candidate entities within acceptable time costs, thereby producing more accurate results.
>
> Reply to W#3:
>
> The similarity in the trends of these lines to some extent is due to the inevitable deterioration in model performance as the proportion of missing facts increases. We enhance the robustness of the rule-based model to this issue by introducing a pre-trained model. As can be seen in Figure 4, the slope of the blue line (INFER) is smaller than the other two lines, especially on ICEWS18, where the slope of the blue line is significantly gentler. This means that the design of our model has mitigated the impact of missing facts on the reasoning results to some extent.
>
> Reply to W#4:
>
> Here we provide an explanation and example regarding Variable Constraints, and we will try our best to include these explanations in the revised version of the PDF.
>
> This is a rule with variable constraints:
>
> $(A, support, C, T)←(A, riot, B, T_1) \bigwedge (B, make statement, A, T_2) \bigwedge (A, resort, C, T_3)$
>
> As we can see in the body of the rule, in the second hop, the entity which B make statement to has to be the same with the entity that riot with B at T1.
>
> If the rule is:
>
> $(A, support, C, T)←(A, riot, B, T_1) \bigwedge (B, make statement, D, T_2) \bigwedge (D, resort, C, T_3)$
>
> Then it is a rule without variable constraints which is different with the rule given above. Because the entity which B make statement to can be or can not be the same with the entity riot with B at T1.
>
> In summary, variable constraints mean that in some rules, it is restricted that the same entity need to appear repetitively so that the rule body is satisfied.
>
> Reply to Q#1:
>
> For facts (s, r, o) that have never appeared, it is meaningless to calculate their Temporal Validity because Temporal Validity measures the validity of a historical fact at the current moment (it should not be considered as a historical fact if it has never appeared). Therefore, we will not update its value in the Temporal Weight Matrices. This means that its corresponding value will be the probability given by the pre-trained model. This process aligns with our motivation to address the bottleneck caused by incompleteness of TKGs. Because the absence of some facts may cause their premises to be invalid, which should have been matched with the bodies of rules.
>
> Reply to Q#2:
>
> Our reply to W#2 can partially address this question. TLogic is limited by the computational cost of path matching, which forces it to stop searching for candidates after a certain number have been found. In contrast, our proposed INFER can efficiently consider a wider range of candidates to obtain more accurate results.
>
> If you are not satisfied with our clarification please feel free to discuss with us.

---

> ### Author Response · Authors · 2024-11-24
> **Looking forward to your reply**
>
> Dear Reviewer cyuQ,
>
> I hope this message finds you well. We are grateful for your recognition of work and sincerely appreciate your valuable and comments. We eagerly anticipate receiving any additional feedback you may have for the clarification we've provided. We totally understand that your time is valuable. Thanks again for your valuable time and efforts and we are looking forward to your response.

---

### Official Review · Reviewer_agps · 2024-11-04

**Soundness:** 2
**Presentation:** 2
**Contribution:** 3
**Rating:** 6
**Confidence:** 4

**Summary:**

The paper introduces INFER, a neural-symbolic approach to temporal knowledge graph extrapolation.  INFER  uses a Temporal Validity Function that captures how frequently facts occur, as well as their validity over time using continuous values. INFER uses pre-trained static knowledge graph embeddings to construct Temporal Weight Matrices. A rule projection module that reformulates rule application as matrix operations.
The paper evaluates INFER's effectiveness using three ICEWS datasets. When tested on modified sparse temporal knowledge graph datasets, INFER shows promising inference capabilities. These results highlight that INFER's combination of continuous temporal validity scoring and GPU-optimized rule application offer useful techniques for temporal knowledge graph reasoning.

**Strengths:**

1. The paper introduces a new scoring mechanism for temporal rule validity.  Alternate techniques in the literature appear to be more complicated.

2. The paper evaluates differentiable rule-based inference systems across several ICEWS datasets, providing a demonstration of these systems on temporal knowledge graphs that represent time in event "timestamp" form.

**Weaknesses:**

1. The author's claimed novelty rests on acceleration of rule-based processing using matrix operations on a GPU.  This is common in differentiable rule-learning systems, apparently first introduced as TensorLog, with associated inductive learning system Neural-LP, see e.g. [1].

2. The paper claims: "Experimental results show that INFER achieves state-of-the-art performance on three datasets and significantly outperforms existing rule-based models on our modified, more sparse TKG datasets, which demonstrates the superiority of our model in inference ability." The authors should consider that both embedding based and rule based systems can perform quite well relative to the methods they compare against.  For instance, consider the following comparison with TimePlex [2]:

| | ICEWS14 | ICEWS14 | ICEWS14 | ICEWS05-15 | ICEWS05-15 | ICEWS05-15 |
|--------|---------|-----|---------|----------|---------|-----|
| Method | MRR | HITS@1 | HITS@10 | MRR | HITS@1 | HITS@10 |
| TimePlex | **60.40** | **51.50** | **77.11** | **63.99** | **54.51** | **81.81** |
| INFER | 44.09 | 34.52 | 62.14 | 48.27 | 37.61 | 68.52 |

The table below illustrates the methods similar to those compared in this paper evaluated on wikidata and yago data sub-sets.  The table also includes a rule-based method (TILP [3]) that is demonstrated to perform on par with Timeplex, illustrating that the performance gap between TimePlex and INFER show above may not be limited to embedding based methods.

| | WIKIDATA12k | WIKIDATA12k | WIKIDATA12k | YAGO11k | YAGO11k | YAGO11k |
|--------|---------|-----|---------|----------|---------|-----|
| Method | MRR | HITS@1 | HITS@10 | MRR | HITS@1 | HITS@10 |
| TLogic | 0.2536 | 0.1754 | 0.4424 | 0.1545 | 0.1180 | 0.2309 |
| ComplEx | 0.2482 | 0.1430 | 0.4890 | 0.1814 | 0.1146 | 0.3111 |
| TA-ComplEx | 0.2278 | 0.1269 | 0.4600 | 0.1524 | 0.0936 | 0.2626 |
| DE-SimplE | 0.2529 | 0.1468 | 0.4905 | 0.1512 | 0.0875 | 0.2674 |
| TimePlex | **0.3335** | 0.2278 | **0.5320** | 0.2364 | **0.1692** | 0.3671 |
| TILP | 0.3328 | **0.2342** | 0.5289 | **0.2411** | 0.1667 | **0.4149** |


[1] @inproceedings{10.5555/3294771.3294992,
author = {Yang, Fan and Yang, Zhilin and Cohen, William W.},
title = {Differentiable learning of logical rules for knowledge base reasoning},
year = {2017},
isbn = {9781510860964},
publisher = {Curran Associates Inc.},
address = {Red Hook, NY, USA},
abstract = {We study the problem of learning probabilistic first-order logical rules for knowledge base reasoning. This learning problem is difficult because it requires learning the parameters in a continuous space as well as the structure in a discrete space. We propose a framework, Neural Logic Programming, that combines the parameter and structure learning of first-order logical rules in an end-to-end differentiable model. This approach is inspired by a recently-developed differentiable logic called TensorLog [5], where inference tasks can be compiled into sequences of differentiable operations. We design a neural controller system that learns to compose these operations. Empirically, our method outperforms prior work on multiple knowledge base benchmark datasets, including Freebase and WikiMovies.},
booktitle = {Proceedings of the 31st International Conference on Neural Information Processing Systems},
pages = {2316–2325},
numpages = {10},
location = {Long Beach, California, USA},
series = {NIPS'17}
}

@inproceedings{jain-etal-2020-temporal,
    title = "{T}emporal {K}nowledge {B}ase {C}ompletion: {N}ew {A}lgorithms and {E}valuation {P}rotocols",
    author = "Jain, Prachi  and
      Rathi, Sushant  and
      {Mausam}  and
      Chakrabarti, Soumen",
    editor = "Webber, Bonnie  and
      Cohn, Trevor  and
      He, Yulan  and
      Liu, Yang",
    booktitle = "Proceedings of the 2020 Conference on Empirical Methods in Natural Language Processing (EMNLP)",
    month = nov,
    year = "2020",
    address = "Online",
    publisher = "Association for Computational Linguistics",
    url = "https://aclanthology.org/2020.emnlp-main.305",
    doi = "10.18653/v1/2020.emnlp-main.305",
    pages = "3733--3747",
    abstract = "Research on temporal knowledge bases, which associate a relational fact (s,r,o) with a validity time period (or time instant), is in its early days. Our work considers predicting missing entities (link prediction) and missing time intervals (time prediction) as joint Temporal Knowledge Base Completion (TKBC) tasks, and presents TIMEPLEX, a novel TKBC method, in which entities, relations and, time are all embedded in a uniform, compatible space. TIMEPLEX exploits the recurrent nature of some facts/events and temporal interactions between pairs of relations, yielding state-of-the-art results on both prediction tasks. We also find that existing TKBC models heavily overestimate link prediction performance due to imperfect evaluation mechanisms. In response, we propose improved TKBC evaluation protocols for both link and time prediction tasks, dealing with subtle issues that arise from the partial overlap of time intervals in gold instances and system predictions.",
}

[3] @inproceedings{xiongtilp,
  title={TILP: Differentiable Learning of Temporal Logical Rules on Knowledge Graphs},
  author={Xiong, Siheng and Yang, Yuan and Fekri, Faramarz and Kerce, James Clayton},
  booktitle={The Eleventh International Conference on Learning Representations}
}

**Questions:**

1. How does your acceleration method differ in nature from TensorLog and Neural-LP?
2. Many methods have developed from the Neural-LP approach.  How does your method compare to those?
3. Why do you not compare to TimePlex as the state of the art method that has been demonstrated on these ICEWS datasets?
4. Can the gap in performance with TimePlex and rule-based sysetsms be explained in terms of the differences between timestamp and interval-time objectives of different temporal knowledge graph methods?

---

> ### Author Response · Authors · 2024-11-16
> **Reply to Reviewer agps**
>
> Thanks for your thorough feedbacks which help us to polish this paper . Here's our clarification and replies to questions in weaknesses:
>
> Reply to W#1 and Q#1:
>
> We have two main motivations(the improvements of rule application efficiency is **not** one of them and it’s more like a advantage of our paradigm instead of the starting point of the design of INFER):
>
> Firstly, traditional rule-based models, when applying rules through path matching, do not consider the validity of historical facts and treat recent facts and ancient facts equally (lines 82-84 in our paper). Secondly, traditional rule-based models are greatly affected by missing information, leading to inference bottlenecks. To address the first issue, we propose a temporal validity function that considers the time span and frequency of historical facts, extending the truth value of facts from binary to continuous and modeling fact validity. The main experiment and ablation experiment verify this motivation. To address the second issue, we introduce a static KG embedding model, which captures structural information through pre-training. We then propose the entire INFER paradigm, which combines embedding information with improved rule application by constructing and updating the Temporal Weight Matrix and proposing Projection operator. The resulting neural-symbolic method gains the ability to reason about missing information and its performance is less affected under incomplete settings compared to traditional rule-based models (Results in Table 2). Meanwhile, the efficiency of applying rules is greatly enhanced through performing matrices operations instead of path matching. More importantly, our model provides a new paradigm to combine embedding-based methods and symbolic-based methods paving the way for future research.
>
> Back to your question, after reading these two papers, we believe that the most fundamental difference between our model and Neural-LP lies in the fact that our proposed method is not learnable and does not require a data-driven training process. Undeniably, our model and TensorLog are indeed similar in essence to a certain extent, and we apologize for neglecting this paper during our research. However, there are still some differences between them. The most significant one is that the relation matrix constructed by TensorLog is **binary**, with values stored as either 0 or 1. Therefore, TensorLog's computational method involves direct matrix multiplication, whose result denotes entities reachable through rules, and is the “OR” of each path. In contrast, our model extends the truth values of events from 0 and 1 to **continuous numerical values**. It is unreasonable to directly compute values through matrix multiplication. Hence, we introduce the Max operation in our computational method.
>
> Reply to W#2:
>
> Thank you for mentioning those two papers. After reading them, we found that the two models you mentioned are targeted at the task of **Interpolation Reasoning** on Temporal Knowledge Graphs (TKGs), while the INFER model we proposed is targeted at the task of **Extrapolation Reasoning** on TKGs. The difference between the two lies in that:
>
> Given a temporal knowledge graph with timestamps varying from $t_0$ to $t_T$, **extrapolation reasoning**  is to predict new links for future timestamps. Given a query with a previously unseen timestamp $(s, r, ?, t) (t>t_T)$, the model need to give a list of object candidates that are most likely to answer the query.  （We give the formal task definition at line 168-170 in our paper）
>
> However under **interpolation settings**, it does not have the future domain restriction and queries are predicted for time t such that $t_0 ≤ t ≤ t_T$.
>
> Furthermore, as we mentioned in lines 364-366 of our paper, to satisfy the requirements of Extrapolation Reasoning, the data in the dataset is sorted by timestamps and then partitioned accordingly.
>
> Therefore, we did not compare INFER with the two models you mentioned, as they are designed for different tasks. I believe this explanation can also answer your questions Q#3 and Q#4.  More details can be found in some early **Extrapolation Reasoning** papers: [1], [2].
>
> [1]Jin W, et al. Recurrent event network: Autoregressive structure inference over temporal knowledge graphs[J]. arXiv preprint arXiv:1904.05530, 2019.
>
> [2] Li Z, Jin X, Li W, et al. Temporal knowledge graph reasoning based on evolutional representation learning[C]//Proceedings of the 44th international ACM SIGIR conference on research and development in information retrieval. 2021: 408-417.

---

> > ### Comment · Reviewer_agps · 2024-11-22
> >
> > I thank the authors for addressing my questions.  I will raise my score.

---

> ### Author Response · Authors · 2024-11-16
> **Reply to Reviewer agps #2**
>
> Reply to Q#2:
>
> As far as we know, no one has ever tried to developed models for TKG extrapolation reasoning based on Neural LP. We are the first to propose a paradigm for integrating rules with embedding methods for TKG extrapolation. There’s one paper published at ACL2024 proposed a machine learning method for rule confidence learning as we mentioned in line229. However, it focuses on improving the rule learning stage while our method mainly improves the rule application stage. As we said in line229, we believe the more accurate confidence given by their model can boost the performance of INFER.
>
> Please feel free to reach out if you have any questions or require further clarification. If you find our response satisfactory, we hope you will consider this a valid reason to consider raising your rating.

---

### Author Response · Authors · 2024-11-28
**General Response**

We would like to thank the reviewers for their constructive feedbacks and we upload our revised PDF based on the discussions we have with all reviewers. Main changes we made in the revised version are (changes are highlighted with brown font):

* Add clarification about Interpolation Reasoning and Extrapolation Reasoning mentioned by Reviewer agps.
* Add examples and illustration for Variable Constraints and clarify the efficiency analysis part suggested by Reviewer cyuQ.
* Add experiments evaluating the performance of INFER using rules with different lengths suggested by Reviewer EcyD.
* Add experiments on various TKG datasets: WIKI and YAGO suggested by Reviewer qjZA.

---

### Public Comment · ~Julia_Gastinger1 · 2025-07-02
**Comment on evaluation protocol: potential issue with "best ranking" tie-break**

I would like to raise a concern about the reported metrics (MRR and Hits@1/3/10). Based on inspecting your public code, it appears your metric computation uses what [Sun et al., 2020] describe as the “best” ranking protocol in the presence of ties.

Specifically, in your repository ([guided_apply.py](https://github.com/JasonLee-22/INFER/blob/main/guided_apply.py)), the evaluation code around line 660 (see attached screenshot with inline annotations) handles ties by assigning the *lowest* (i.e. best) possible rank to the ground-truth entity if multiple entities share the same model score:

```
ranking = (sort_score[i, :] == target_score[i]).nonzero()[0]
ranking = 1 + ranking.item()
```
This line selects the indices where the sorted score list matches the score of the ground-truth node, and then takes the first such index — i.e., the smallest one. As a result, the evaluation always assigns the best possible rank to the ground-truth entity in the case of ties, rather than using an average or random tie-breaking strategy..

This protocol is known to inflate evaluation metrics unfairly, since it does not reflect the true ranking uncertainty in the presence of score ties. Sun et al. recommend fairer alternatives (such as assigning the average rank over tied entities).

The reported results in the paper appear to rely on this tie-breaking strategy. I therefore strongly suggest re-running evaluation with a fairer tie-breaking method (e.g., average or random ranking) to ensure valid and comparable metrics.

[Sun et al., 2020] Zhiqing Sun, Shikhar Vashishth, Soumya Sanyal, Partha P. Talukdar, and Yiming Yang. A reevaluation of knowledge graph completion methods. In Proceedings of the 58th Annual Meeting of the Association for Computational Linguistics (ACL), pages 5516–5522, 2020.

---

> ### Public Comment · ~Ningyuan_Li2 · 2025-07-02
> **Reply to Julia Gastinger**
>
> Hi Julia,
>
> Thank you for your interest in our work! I truly appreciate that you’ve taken the time to thoroughly inspect our code.
>
> Regarding the issue you raised in your public comment: yes, we do use the "best" setting for evaluation. I recommend checking the public code provided by TLogic [1] (https://github.com/liu-yushan/TLogic/tree/main, particularly the evaluate.py file). TLogic was the first to leverage rule-based approaches for TKG extrapolation reasoning, and in their released code, they also report results under the "best" setting for comparison with other embedding-based methods.
>
> For fair comparison and alignment, that's why we adopted the same "best" setting. Your concern is absolutely valid—this setting may introduce some bias. However, we believe that if all rule-based methods are evaluated under the same "best" settings, the comparison remains relatively fair. If embedding-based methods are involved then there might be some bias.
>
> Thus if it does not convince you, you can try to reproduce the results reported by TLogic and other several rule-based methods mentioned in this paper and then carry out your further research. I think the code given by TLogic provides the calculation for other settings("average", "worse").
>
> [1] Liu, Y., Ma, Y., Hildebrandt, M., Joblin, M., & Tresp, V. (2022, June). Tlogic: Temporal logical rules for explainable link forecasting on temporal knowledge graphs. In Proceedings of the AAAI conference on artificial intelligence (Vol. 36, No. 4, pp. 4120-4127).

---

> > ### Public Comment · ~Christian_Meilicke1 · 2025-07-02
> >
> > Dear Ningyuan,
> >
> > I would alos like to add something to this point. Some years ago, I got aware of this IJCAI paper on relational learning.
> > Dumancic, Garcia-Duran, Niepert: A comparative study of distributional and symbolic paradigms for relational learning. IJCAI 2019
> > https://www.ijcai.org/proceedings/2019/0843.pdf
> >
> > It compares symbolic approaches (rule learning) against non-symbolic approaches with respect to classification and knowledge base completion (in the non temporal setting).
> >
> > I would like to point your attention to Table 2 on page 6092. Here you can see the results of the classic rule learner TILDE if applied to some of the standard knowledge base completion tasks FB237 and WN18RR.
> >
> > Especially the results for FB237 are superior to any other method. These results are so far above the results of other models that I had to doubt their correctness (and everbody who knows the dataset and other results should also be at least … surprised).
> >
> > After checking the code I found that the the evaluation protocoll was the “best” ranking protocol. The same protocoll that you used for evaluating INFER.
> > If a rule-based approach (or any other approach) assigns the same scores to different candidates, this can have a very strong impact on the MRR. Especially a rule-based approach can tend to do this, if you have, fro example only few rules.  I checked the rules and the rankings and found that this is the case. There there were lots of ties. I informed the author, and he was so kind to rerun all experiments with the corrected evaluation protocoll.
> >
> > The new results were completely different. You can find them here, in the final version v4 of the paper.
> >
> > https://arxiv.org/pdf/1806.11391v4
> > See again Table 2. The HitsAT1 went down from 0.763 to 0.12. Similar for the other numbers in this table. Not knowing this, when the IJCAI paper was accepted, the authors wrote this at the end as one of three important results:
> >
> > “Third, the symbolic methods outperform the distributional ones on the task of knowledge base completion.”
> >
> > This conclusion was completely wrong.
> >
> > I do not think that the authors of the IJCAI paper did this on purpose. Moreover, it was great that they corrected their results and made an updated version of the paper available.
> > I also do not think that you have chosen this evaluation protocoll to gain any kind of advantage. However, I think that you underesteimate the potentiel impact of this protocoll and its difference to the average or random protocoll. The story I told about this IJCAI paper might illustrate this.
> >
> > Best regards, Christian.

---

> > ### Public Comment · ~Julia_Gastinger1 · 2025-07-02
> >
> > Hello Ningyuan,
> >
> > Thank you for your prompt reply and for the clarification.
> >
> > In previous works, we studied different evaluation setups and inconsistencies for TKG methods, and in this context re-evaluated (amongst others) both TLogic and TRKG (both are rule-based methods). For our evaluation, we used the **"random"** setting for tie-breaking.
> >
> > We observed that especially for TRKG, the results changed significantly compared to the original paper.
> >
> > Perhaps these findings might be helpful for your further evaluation efforts.
> >
> > Here are the links:
> >
> > - **Study on evaluation settings and re-evaluation of TLogic**: [Comparing Apples and Oranges? On the Evaluation of Methods for Temporal Knowledge Graphs](https://github.com/JuliaGast/JuliaGast.github.io/blob/master/files/gastinger_evaluation_paper_TKG.pdf)
> > - **GitHub repository**: [TKG-Forecasting-Evaluation](https://github.com/nec-research/TKG-Forecasting-Evaluation/tree/main?tab=readme-ov-file)
> > - **Results for TRKG**: [History Repeats Itself: A Baseline for Temporal Knowledge Graph Forecasting](https://arxiv.org/abs/2404.16726)
> >
> > Kind regards,
> > Julia

---

### Meta-Review · Area_Chair_GXN5 · 2024-12-24

**Metareview:**

The proposed method INFER offers a promising approach to temporal knowledge graph reasoning. The majority of the reviewers’ concerns have been addressed, and the authors have provided sufficient clarifications, additional experiments, and analysis to strengthen the manuscript. Given the promising results, the addressed concerns, and the contributions made by the authors, I recommend accepting the paper.

**Additional Comments On Reviewer Discussion:**

The authors have made significant revisions to their manuscript based on the feedback provided by the reviewers. The following changes were made in response to the reviewers' concerns:

Clarification on Interpolation and Extrapolation Reasoning: The authors provided additional explanations on the Interpolation and Extrapolation Reasoning concepts, as suggested by Reviewer agps.

Clarification on Variable Constraints and Efficiency: The authors have added examples and illustrations for Variable Constraints, as well as expanded their explanation of the efficiency analysis, addressing Reviewer cyuQ's concerns.

Experiments on Rule Lengths: The authors included experiments evaluating the performance of INFER using rules with different lengths, as suggested by Reviewer EcyD.

New Datasets: The authors extended their experiments to include additional TKG datasets, such as WIKI and YAGO, as recommended by Reviewer qiZA.

After the rebuttal, two reviewers raised their scores. Although the reviewers did not reach a consensus on accepting this paper, all agreed that the novelty of the approach is solid. This is why I recommend accepting the paper.

---

### Decision · Program_Chairs · 2025-01-22

Accept (Poster)